# Fatty Acids Composition and HIV Infection: Altered Levels of n-6 Polyunsaturated Fatty Acids Are Associated with Disease Progression

**DOI:** 10.3390/v15071613

**Published:** 2023-07-23

**Authors:** Thor Ueland, Bjørn Waagsbø, Rolf K. Berge, Marius Trøseid, Pål Aukrust, Jan K. Damås

**Affiliations:** 1Research Institute of Internal Medicine, Oslo University Hospital, Rikshospitalet, NO-0420 Oslo, Norway; 2Faculty of Medicine, University of Oslo, NO-0318 Oslo, Norway; 3Thrombosis Research Center (TREC), Division of Internal Medicine, University Hospital of North Norway, NO-9038 Tromsø, Norway; 4Centre of Molecular Inflammation Research, Department of Clinical and Molecular Medicine, Norwegian University of Science and Technology, NO-7491 Trondheim, Norway; 5Department of Infectious Diseases, St Olav’s Hospital, NO-7006 Trondheim, Norway; 6Department of Clinical Science, University of Bergen, NO-5021 Bergen, Norway; 7Department of Heart Disease, Haukeland University Hospital, NO-5021 Bergen, Norway; 8Section of Clinical Immunology and Infectious Diseases, Oslo University Hospital Rikshospitalet, NO-0424 Oslo, Norway

**Keywords:** arachidonic acid, fatty acids, HIV, monocytes, mycobacteria

## Abstract

Fatty acids (FAs) are important regulators of immune responses and innate defense mechanisms. We hypothesized that disturbed FA metabolism could contribute to the progression of HIV infection. Plasma levels of 45 FAs were analyzed with gas chromatography in healthy controls and HIV-infected patients with regard to Mycobacterium avium complex (MAC) infection. In vitro, we assessed MAC-PPD-induced release of inflammatory cytokines in peripheral and bone marrow mononuclear cells (PBMC and BMMC) according to levels of n-6 polyunsaturated fatty acids (PUFAs). While plasma saturated FAs were higher in HIV infection, PUFAs, and in particular the n-6 PUFA arachidonic acid (AA), were lower in patients with advanced disease. The ratio between AA and precursor dihomo-γ-linolenic acid, reflecting Δ5-desaturase activity, was markedly lower and inversely correlated with plasma HIV RNA levels in these patients. Depletion of AA was observed prior to MAC infection, and MAC-PPD-induced release of TNF and IL-6 in PBMC and BMMC was lower in patients with low plasma AA. Our findings suggest that dysregulated metabolism of n-6 PUFAs may play a role in the progression of HIV infection. While high AA may contribute to chronic inflammation in asymptomatic HIV-infected patients, low AA seems to increase the susceptibility to MAC infection in patients with advanced disease.

## 1. Introduction

Progression of HIV infection is frequently associated with multiple metabolic and cardiovascular complications [1,2]. These complications could be related to side effects of anti-retroviral therapy (ART), but HIV infection per se, even during successful ART, may also induce metabolic alternations through mechanisms such as chronic low-grade systemic inflammation [3,4].

Fatty acids (FAs), both free and as part of other lipids (i.e., triglycerides, phospholipids, sphingolipids and sterol lipids), play a number of key roles in cellular energy metabolism and are essential cell membrane components. FAs also have the ability to signal through receptors such as peroxisome proliferator-activated receptors and liver X receptors [5,6], and disturbed FA composition has been suggested to influence the development of various metabolic, cardiovascular and inflammatory disorders [7,8]. Thus, some polyunsaturated FAs (PUFAs), including arachidonic acid (AA; C20:4n-6), eicosapentaenoic acid (EPA; C20:5n-3) and docosahexaenoic acid (DHA; C22:6n-3), can serve as precursors for the synthesis of bioactive lipid mediators (e.g., prostaglandins [PG], leukotrienes [LT], lipoxins and resolvins) [9,10], mediating both inflammatory and anti-inflammatory effects [10,11,12]. There has also been focused attention on the central role of FAs in immune cell regulation, highlighting that FAs have a direct influence on a number of cellular processes involved in T-cell responses and antigen presentation [13,14,15,16].

There is some evidence indicating that FA metabolism is disturbed in HIV-infected patients. Moreover, a recent meta-analysis suggested that n-3 PUFA supplementation may attenuate inflammation in HIV-infected patients, as assessed by decreased levels of C-reactive protein (CRP) also in those that received ART [17]. In an earlier meta-analysis, it was shown that n-3 PUFA supplementation down-regulated triglyceride levels in HIV-infected patients on ART [18], and this was confirmed in a more recent meta-analysis [19]. Most reports have focused on abnormalities in plasma levels of FAs in HIV-infected patients treated with protease inhibitors and in patients with lipodystrophy syndrome [20,21,22,23]. A few studies, however, have shown alterations in FA composition in untreated HIV-infected children and during early HIV infection, potentially contributing to disease progression in individuals [24,25,26]. In support of this notion, there is accumulating evidence that FAs play a crucial role in the innate immune responses against various infections such as influenza and tuberculosis [27,28]. Moreover, several studies have shown that n-3 PUFA may suppress prototypical inflammatory cytokines like tumor necrosis factor (TNF), interleukin (IL)-1 and IL-6 [9,29,30], all with relevance for HIV-related pathology [31]. There are also some experimental data from the simian immunodeficiency virus-infected macaque model showing that FAs may be involved in the progression of retrovirus-induced immunodeficiency [30], and it has been suggested that cellular FA synthase is required for late stages of HIV-1 replication [32]. However, the FA composition in untreated adult HIV-infected patients is still not fully clarified. Moreover, whereas most studies have focused on n-3 PUFA, data on n-6 and n-9 PUFA are scarcer.

Based on their role in inflammation and immunity regulation, we hypothesized that dysregulated FA metabolism may be involved in disease progression in HIV-infected patients. To evaluate this hypothesis, we determined levels of a wide range of FAs in plasma from HIV-infected patients with various disease severities including patients classified as rapid clinical progressors and patients with advanced HIV-related disease acquiring Mycobacterium avium complex (MAC) infection. We also examined if ART influenced the FA pattern in HIV-infected patients.

## 2. Materials and Methods

### 2.1. Patients and Controls

A Flow Chart of the Different Cohorts and Substudies Are Given in Figure 1.

Cross-sectional study: 60 HIV-infected patients were included (34 ± 6 years, 38 men and 22 women) in the period 1990–2005. Sampling was performed before initiation of ART. 20 patients were classified in Centers for Disease Control and Prevention (CDC) class A (CD4+ T cell count, 465 ± 50 × 10^6^/L; HIV RNA level, 101 ± 35 × 10^3^ copies/mL plasma), 22 in CDC class B (CD4+ T cell count, 198 ± 35 × 10^6^/L; HIV RNA level, 155 ± 76 × 10^3^ copies/mL plasma) and 18 in CDC class C (CD4+ T-cell count, 45 ± 17 × 10^6^/L; HIV RNA level, 935 ± 54 × 10^3^ copies/mL plasma). We also included 20 sex- and age-matched healthy controls.

Longitudinal study: Of the 60 HIV-infected patients, plasma was available for 36 months before inclusion in the cross-sectional study in 19 patients. Blood samples in this sub-study were collected before ART was available in Norway (June 1996). Of the 19 patients, 9 were categorized as rapid clinical progressors and 10 as non-progressors. Rapid progressors satisfied one of these criteria during follow-up: death due to HIV-related complications; clinical progression as reflected by altered CDC classification; a new AIDS-defining event.

ART study: In a subgroup of patients (*n* = 30) followed during ART, plasma samples were taken before (baseline) initiation of treatment with nucleoside/nucleotide analogs in combination with protease inhibitors (PI)/ritonavir (r) (*n* = 15) and integrase inhibitors (*n* = 15) and 3, 6 and 12 months after. 

MAC study: We included 31 patients with AIDS and MAC infection (35 ± 8 years, 23 males and 8 females,). Plasma samples were isolated 3–12 months before diagnosis and at diagnosis. Twenty-eight patients with AIDS but without MAC infection (36 ± 9 years, 22 males and 6 females) had temporal assessment during the same period. All patients with MAC infection had one blood culture positive for MAC at minimum. The AIDS groups were matched for age, ART (i.e., zidovudine and lamivudine) and CD4+ T-cell counts (<50 × 10^6^/L). 

Informed consent for blood sampling was obtained from all subjects. The studies were approved by the local ethical committee; ethics committee name: REK Sør, approval code: S07164b (2012/521), approval date: 26 April 2007.

### 2.2. Blood Sampling Protocol and Routine Analyses

Fasting blood samples were collected into EDTA-containing chilled tubes (Becton Dickinson, Franklin Lakes, NJ, USA), placed on ice and centrifuged within 20 min (2000 g for 20 min) to obtain platelet-poor plasma, aliquoted and stored at −80 °C. HIV RNA was measured using quantitative reverse polymerase chain reaction (detection limit 40 copies mL^−1^; Amplicor HIV Monitor; Roche Diagnostic Systems, Branchburg, NJ, USA).

### 2.3. Measurements of Plasma Levels of FAs

Measurement of total levels and composition of FAs was done following extraction of lipids from plasma using a mixture of methanol and chloroform. Extracts were transesterified using H_2_SO_4_/methanol. Recovered fatty acid methyl esters were quantified using gas chromatography as previously described [33]. Each FA was expressed as percentage of total FAs. For overview of FA see Table 1.

### 2.4. Peripheral Blood Mononuclear Cells (PBMC) Culture

PBMC was obtained from 10 healthy controls and 20 HIV-infected patients using density gradient centrifugation (Lymphoprep; Nycomed Pharma, Oslo, Norway) of heparinized blood, further incubated in 96-well plates (flat-bottomed, 2 × 10^6^/mL; Costar, Cambridge, MA, USA) in medium (RPMI 1640 with 2 mM L-glutamine and 25 mM HEPES buffer [PAA Laboratories, Pasching, Austria] supplemented with 10% fetal calf serum [PAA Laboratories]) or stimulated with phytohaemagglutinin (PHA 1:200; Murex, Dartford, UK) and purified protein derivate (PPD) of MAC (MAC-PPD; 1:100). Cell-free supernatants were collected and stored at −80 °C after culturing for 20 h.

### 2.5. Bone Marrow Mononuclear Cells (BMMC) Culture

Heparinized bone marrow samples from 9 HIV-infected patients and 5 healthy controls were obtained by aspiration from the posterior iliac crest, and BMMC were prepared by density gradient centrifugation (Lymphoprep; Nycomed Pharma). BMMC were incubated in flat-bottomed 96-well trays (2 × 10^6^/mL; Costar), in medium alone [RPMI-1640 with 2 mM L-glutamine and 25 mM HEPES buffer (Gibco, Paisley, UK) supplemented with 10% FCS (Sigma, St Louis, MO, USA)] or stimulated with PHA (final dilution 1:100) and MAC-PPD (final dilution 1:100).

### 2.6. Analysis of Circulating Lipopolysaccharide (LPS) 

LPS was analyzed using limulus amebocyte lysate colorimetric assay (Lonza, Walkersville, MD, USA) according to the manufacturer’s instructions.

### 2.7. Enzyme Immunoassays (EIAs)

Concentrations of TNF, interferon (IFN)γ and IL-6 were measured in duplicate using EIA obtained from R&D Systems (Minneapolis, MN, USA).

### 2.8. Statistical Analysis

For the cross-sectional study comparing FA composition in HIV-infected patients without ART and healthy controls, we used the Mann–Whitney U test. For the cross-sectional study when comparing FA composition in HIV-infected patients according to CDC classification, the Kruskal–Wallis was used à priori, followed by Mann–Whitney U test to determine differences between groups. For analyzing temporal data in the longitudinal study (i.e., in progressors vs. non-progressors), namely the ART study (i.e., in patients receiving ART, comparing patients with high and low baseline arachidonic acid) and the MAC study (i.e., comparing patients with AIDS with or without MAC infection), paired differences within groups were first assessed with the Friedman test for repeated measures followed by the Wilcoxon signed-rank test for evaluating each time point with baseline while between-group differences were compared using the Mann–Whitney U test. Correlations were assessed with Spearman’s rank test.

For in vitro data, cytokine levels during stimulation (i.e., PHA or MAC-PPD) were compared to unstimulated cells using the Wilcoxon signed-rank test while differences between groups (i.e., controls and patients with high or low baseline arachidonic acid) were compared with the Kruskal–Wallis test followed by the Mann–Whitney U test to determine the differences between groups.

In the text, the data are given as mean ± SEM. Probabilities are 2-sided and considered to be significant when less than 0.05.

## 3. Results

### 3.1. Cross-Sectional Study

#### 3.1.1. Plasma Levels of FAs in Untreated HIV-Infected Patients and Controls 

In the cross-sectional analysis of 60 HIV-infected patients without ART and 20 healthy controls, plasma FA composition differed significantly between HIV-infected patients and controls. While the overall plasma levels of the saturated FAs (SFAs) and monounsaturated FAs (MUFAs) were modestly higher in HIV-infected patients irrespective of disease severity, the PUFAs were lower in these patients (Figure 2). Both n-3 PUFAs, including EPA (C20:5n-3) and DHA (C22:6n-3), n-6 PUFAs and n-9 PUFAs were lower in HIV-infected patients compared to controls and were all significantly associated with advanced clinical disease (Figure 2A). However, significant differences between the CDC classes were only observed for the n-6 PUFA levels. We, therefore, focused on the n-6 PUFAs in further analyses.

#### 3.1.2. n-6 PUFA Pattern in HIV-Infected Patients

Among the n-6 PUFAs, plasma levels of AA (C20:4n-6) were particularly low in HIV-infected patients with advanced disease (Figure 2B). In contrast, higher levels of AA were observed in patients with asymptomatic disease (i.e., CDC class A). A different pattern was observed for the precursors of AA. Thus, plasma levels of the immediate precursor for AA, dihomo-γ-linolenic acid (DGLA; C20:3n-6), were higher in HIV-infected patients, with particularly high levels in symptomatic disease (i.e., CDC class B and C), while the other precursors of AA, linoleic acid (LA; C18:2n-6) and γ-linolenic acid (GLA; C18:3n-6) were modestly lower in HIV-infected patients with no differences between the CDC classes (Figure 2B). The further elongated forms of AA, docosatetraenoic acid (DTA; C22:4n-6) and docosapentaenoic acid (DPA; C22:5n-6) were also markedly lower in HIV-infected patients with advanced disease, suggesting a reduced pool of AA in these patients (Figure 2B).

As the metabolism of LA involves several desaturases and elongates, the ratios between the different LA metabolites may give an estimate of endogenous enzyme activities. While the GLA/LA and DPA/DTA ratios (reflecting Δ6-desaturase and Δ4-desaturase activity, respectively) were similar in HIV-infected patients and controls, the AA/DGLA-ratio (reflecting Δ5-desaturase activity) was 41% lower in HIV-infected patients, with particularly low levels in AIDS patients (i.e., CDC class C), suggesting a profoundly reduced Δ5-desaturase activity in patients with advanced HIV infection.

#### 3.1.3. Relation between ω-6 PUFAs and HIV RNA, CD4 T Cell Counts, Inflammatory Cytokines and Endotoxin Levels

We found a significant inverse correlation between HIV RNA (copies/mL) and levels of AA (r = −0.52, *p* < 0.01), DTA (r = −0.59, *p* < 0.01) and DPA (r = −0.65, *p* < 0.05), *p* < 0.01) and significant positive correlation with DGLA (r = 0.42, *p* < 0.05). Furthermore, HIV RNA showed a strong inverse correlation with the AA/DGLA-ratio (r = −0.65, *p* < 0.01), suggesting an interaction between HIV and Δ5-desaturase activity. Plasma levels of TNF, but not the other cytokines, were correlated with levels of AA (r = −0.61, *p* < 0.01), DTA (r = −0.61, *p* < 0.01), DPA (r = −0.59, *p* < 0.05) and DGLA (r = 0.47, *p* < 0.05). As the translocation of endotoxins could influence FA metabolism and inflammation in HIV-infected patients, we measured plasma levels of LPS in this study population. Plasma levels of LPS were correlated with the levels of AA (r = −0.64, *p* < 0.01), DTA (r = −0.61, *p* < 0.01), DPA (r = −0.59, *p* < 0.05) and DGLA (r = 0.62, *p* < 0.01) while no correlations were found with LA and GLA. 

### 3.2. Longitudinal Study: n-6 PUFAs in Rapid Progressors and Long-Term Non-Progressors

We next examined the levels of these FAs during disease progression before initiating ART. While the plasma levels of AA did not change significantly during the early phase of progression, a rapid decline in the AA levels was observed during progression at the advanced stages of disease. In contrast, DGLA increased in rapid progressors at the end of the 3-year follow-up (*n* = 9) (Figure 2C). As for non-progressors, minimal changes were observed over time for both AA and DGLA (*n* = 10) (Figure 2C). Findings similar to those for AA were observed for DTA and DPA (Figure 2C).

### 3.3. ART Study: Plasma Levels of n-6 PUFAs during ART

We then examined the levels of FAs in 22 HIV-infected patients during ART. As shown in Figure 2D, while plasma levels of AA increased and levels of DGLA declined during ART in patients with low baseline AA levels (<5.0 wt%), no changes were observed in patients with high levels of AA before treatment (>7.0 wt%). Findings similar to those for AA were observed for DTA and DPA. In contrast, plasma levels of LA and GLA did not change during ART in either of the groups (Figure 2D). There were no significant differences between the different ART regimens (i.e., protease inhibitors versus integrase inhibitors) for either AA or DGLA. Although not fully normalized, the rise in AA and the decline in DGLA during ART in those with low AA levels before initiating may be regarded as a potential beneficial effect of ART. Notably, in these patients, but not in the ART group as a whole, we found significant correlations between the change of AA (r = −0.63, *p* < 0.01) and DGLA (r = 0.45, *p* < 0.05) and the change in HIV RNA during ART. We found no association between changes in AA or DGLA levels and changes in CD4+ T-cell counts during ART.

### 3.4. MAC Study

#### 3.4.1. Plasma Levels of n-6 PUFAs in Patients Acquiring MAC Infection

To further examine the regulation of FAs in advanced disease, we assessed plasma n-6 PUFAs in HIV-infected patients with CD4+ T cell count < 50 cells/µL at various time points during follow-up (*n* = 59). A comparison of the plasma levels of n-6 PUFAs was made between patients who acquired MAC infection (*n* = 31) and those who did not (*n* = 28). We found a significant depletion of AA and an accumulation of DGLA prior to MAC infection (Figure 3A). Findings similar to those for AA were observed for DTA and DPA (Figure 3). In contrast, no corresponding changes were seen in HIV-infected patients who did not acquire MAC infection (Figure 3A).

#### 3.4.2. Cytokine Production in PBMCs from HIV-Infected Patients with Either High or Low Plasma Levels of AA

To elucidate potential functional consequences of the decreased levels of AA during HIV infection on innate immune mechanisms, we analyzed the MAC-PPD induced release of TNF, IFNγ and IL-6 in PBMC and BMMC from patients with either high or low plasma levels of AA before initiating ART. As shown in Figure 3B,C, the release of TNF, IFNγ and IL-6 was significantly increased in PHA- and MAC-PPD-stimulated PBMC and BMMC from patients with high AA levels. However, in HIV-infected patients with low levels of AA, the release of TNF in PBMC (Figure 3B) and TNF and IL-6 in BMMC (Figure 3C) was significantly decreased in both PHA- and MAC-PPD-stimulated PBMC comparing patients with high AA levels (Figure 3B).

## 4. Discussion

In the present study, we performed a thorough analysis of PUFAs, and in particular n-6 PUFAS, in relation to HIV infection, and a brief description of the most important PUFAs in this article are given in Table 1. Together, our findings suggest a close association between disturbed n-6 PUFA metabolism, disease progression, HIV replication and reduced host defense against MAC infection in HIV-infected patients. These disturbances were, to some degree, modulated by ART, with no differences between integrase- and protease-containing regimens but notably, without full normalization compared to healthy controls.

There are some previous studies that have shown disturbed plasma levels of n-6 PUFAs in HIV infection [20,24,26]. In particular, these studies have shown lower plasma levels of LA (C18:2n-6) in HIV-infected patients with advanced disease [24]. Interestingly, a study in HIV-infected patients in Uganda showed that the risk of death or hospitalization decreased significantly with an increase in LA or GLA levels, and low levels of LA and GLA were associated with low CD4+ T cell counts [25]. LA is an essential FA because humans cannot synthesize it. Decreased LA plasma levels in HIV-infected patients have therefore been linked to malnutrition and reduced dietary intake in HIV-infected patients [34,35]. Other studies have shown disturbed levels of LA after exposure to ART and in patients with fat redistribution after long-term ART [20,21,22]. We have no detailed information on the dietary habits of our HIV cohorts. However, we found no association between the marginally reduced plasma levels of LA and bodyweight or lipoproteins in our study, nor were there any associations between LA levels and exposure to ART.

In contrast to the modest findings on LA and GLA, we found markedly disturbed AA levels. Whereas AA was higher in asymptomatic patients, it was markedly lower in advanced HIV infection. Based on the proposed pro-inflammatory role of AA, these findings may seem conflicting but could possibly reflect a dual role for AA in HIV progression. Thus, asymptomatic and even fully viral-suppressed HIV-infected patients are characterized by chronic low-grade inflammation [2,3,4], and enhanced AA levels in these patients could contribute to inflammatory processes via several direct and indirect mechanisms [12,36]. However, while high levels of AA could contribute to low-grade systemic inflammation, low AA levels could impair the ability to promote an inflammatory response when needed. Herein we found a significant depletion of AA preceding MAC infection in patients with advanced disease. In these individuals, the CD4+ T-cell counts were comparable in patients acquiring and not acquiring MAC, suggesting that AA levels may influence T cell or other PBMC subset function rather than merely cell depletion. In support of this, we show decreased release of TNF after MAC- and PHA-stimulation of PBMC from HIV-infected patients with low levels of AA. Thus, low AA levels could contribute to impaired inflammatory responses when challenged by microbes such as MAC in advanced HIV-related disease [27,37,38,39].

Although we cannot exclude that lifestyle factors could have potentially contributed to the altered n-6 PUFA levels in our study, there is increasing evidence for a direct influence of HIV itself on PUFA metabolism [24,25,26]. A direct role for HIV on n-6 PUFA metabolism is also supported by our findings of significant correlations between HIV RNA and AA and DGLA levels in untreated HIV-infected patients as well as the correlations between changes in these parameters during progression of the disease. Moreover, we also found an association between HIV RNA and a markedly decreased AA/DGLA ratio, indicating a profoundly attenuated Δ5-desaturase activity and suggesting a direct interaction between HIV and altered AA generation. Rasheed et al. have shown that HIV replication alone (without any influence of antiviral drugs) can affect cellular enzymes and proteins that are involved in lipid synthesis, transport and metabolism, also supporting a direct effect of HIV on n-6 PUFA metabolism [40]. However, ART did not restore AA or DGLA levels to normal levels, even in fully virally suppressed patients, suggesting that factors other than HIV viremia contribute to reduced levels of AA in HIV-infected patients such as persistent low-grade inflammation that may be seen persisting even during successful ART.

Our findings of significant associations between several n-6 PUFAs and plasma levels of TNF and LPS suggest that chronic low-grade systemic inflammation and microbial translocation from the gastrointestinal tract could influence these disturbances of n-6 PUFAs in HIV-infected individuals, possibly contributing to the residual disturbances of n-6 PUFA levels after suppression of HIV [3,41]. Moreover, the increase in DGLA and decrease in AA during advanced HIV infection was also correlated with levels of TNF and LPS, suggesting that inflammation, potentially reflecting gut-leakage mechanisms, could influence Δ5-desaturase activity. Further studies on the interaction between HIV replication within gut-associated lymphoid tissue, LPS translocation, systemic inflammation and n-6 PUFA metabolism are therefore warranted. Recent studies suggest that altered gut microbiota including translocation of microbial products contribute to HIV-related low-grade systemic inflammation and metabolic disturbances [42], and the present study further support such a notion by showing a strong correlation between LPS levels and disturbed n-6 PUFA levels.

So far, measurements of PUFAs are not part of the routine analyses at most hospitals; therefore, until such analyses are more available on a routine basis, we will not recommend using PUFA measurements in the routine management of people living with HIV. However, supplemental measurement of PUFAs, not only n-3 PUFAs but also n-6 PUFAs, could be considered at least in subgroups such as those with metabolic and cardiovascular co-morbidities. Most importantly, however, we need large randomized, controlled trials to evaluate if such supplementation could modulate the low-grade inflammation and immune activation that are seen in people living with HIV even during successful ART.

The present study has some limitations. The number of patients, and in particular of healthy controls, were low. However, several correlation analyses were performed; nevertheless, some correlations could be by chance and importantly, correlations do not necessarily mean any causal relationships.

## 5. Conclusions

In conclusion, we found a profound dysregulation of n-6 PUFA in HIV-infected patients, particularly in those with advanced disease. Our study indicates a direct inhibition of Δ5-desaturase activity by HIV, leading to high levels of DGLA and low levels of AA in untreated HIV-infected patients and possibly reducing the defense against opportunistic mycobacterial infections in patients with advanced HIV infection. Although most of the samples were from untreated patients, our data also suggest persistent pathology with elevated AA levels, even under successful ART with full viral suppression, suggesting that targeted therapy against these n-6 disturbances should be further explored in people living with HIV.

## Figures and Tables

**Figure 1 viruses-15-01613-f001:**
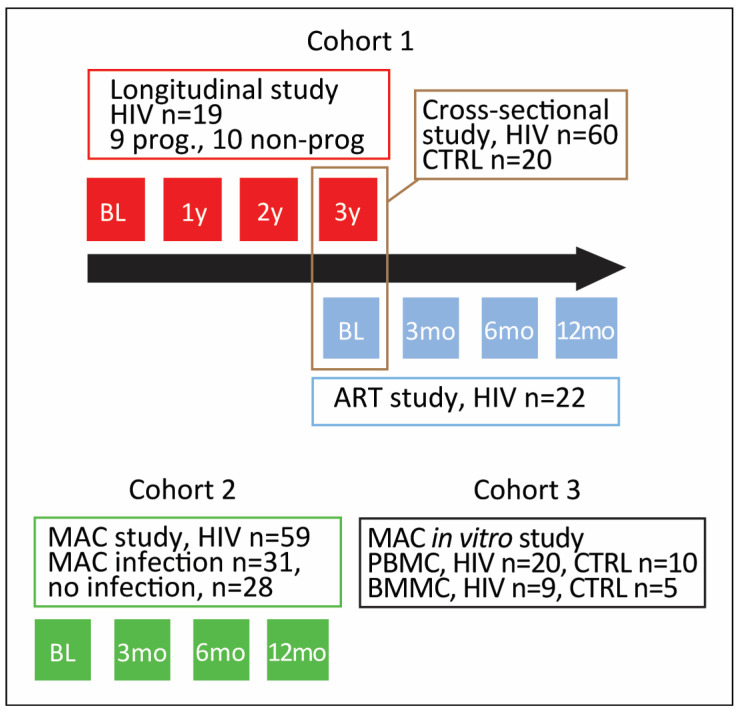
Overview of cohorts used in the study. BL, baseline; y, year; prog, progressors; PBMC, peripheral blood mononuclear cells; BMMC, bone marrow mononuclear cells.

**Figure 2 viruses-15-01613-f002:**
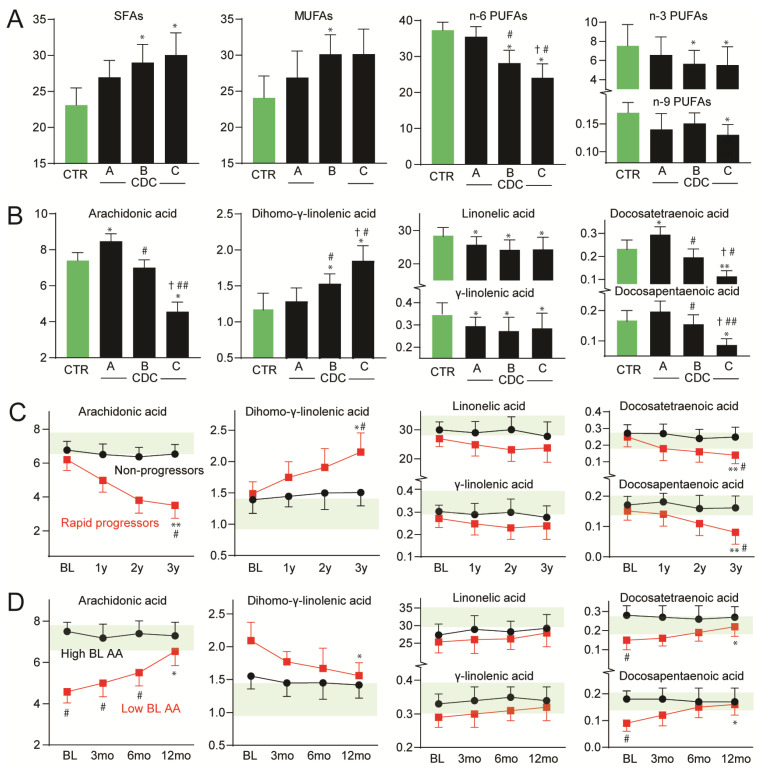
Plasma levels of FAs in HIV-infected patients and controls and according to disease progression and ART treatment. Plasma levels of (**A**) saturated fatty acids (SFAs), monounsaturated fatty acids (MUFAs), n-3/6/9 polyunsaturated fatty acids (PUFAs); (**B**) n-6 PUFAs arachidonic acid, dihomo-γ-linolenic acid, linoleic acid, γ- linolenic acid, docosatetraenoic acid and docosapentaenoic acid in 60 HIV-infected patients without antiretroviral therapy (ART), classified into Centers for Disease Control and Prevention (CDC) class A (*n* = 20, asymptomatic HIV infection), CDC class B (*n* = 22, symptomatic non-AIDS) and in CDC class C (*n* = 18, AIDS) and in healthy controls (*n* = 20). For A and B, Kruskal–Wallis was used à priori, followed by Mann–Whitney U test to determine differences between groups. * *p* < 0.05, ** *p* < 0.01 vs. controls, # *p* < 0.05, ## *p* < 0.01 vs. CDC class A, † *p* < 0.05 vs. CDC class B. Plasma levels of n-6 PUFAs during (**C**) longitudinal testing in HIV-infected patients categorized as rapid clinical progressors (*n* = 9) and non-progressors (*n* = 10) given as years (y) after baseline (mean observation time 43 months) and (**D**) during antiretroviral therapy in HIV-infected patients with high (>7.0 wt%, *n* = 14) and low (<5.0 wt%, *n* = 8) baseline plasma levels of arachidonic acid (AA), given as months (mo) after initiation of therapy. For (**C**,**D**), paired differences within groups were first assessed with the Friedman test for repeated measures followed by the Wilcoxon signed-rank test for evaluating each time point with regard to baseline while between-group differences were compared using the Mann–Whitney U test. * *p* < 0.05, ** *p* < 0.01 vs. baseline, # *p* < 0.05 between groups at the same time-point. Dates are shown as mean ± SD. Green shading indicated the mean ± SD levels in healthy controls.

**Figure 3 viruses-15-01613-f003:**
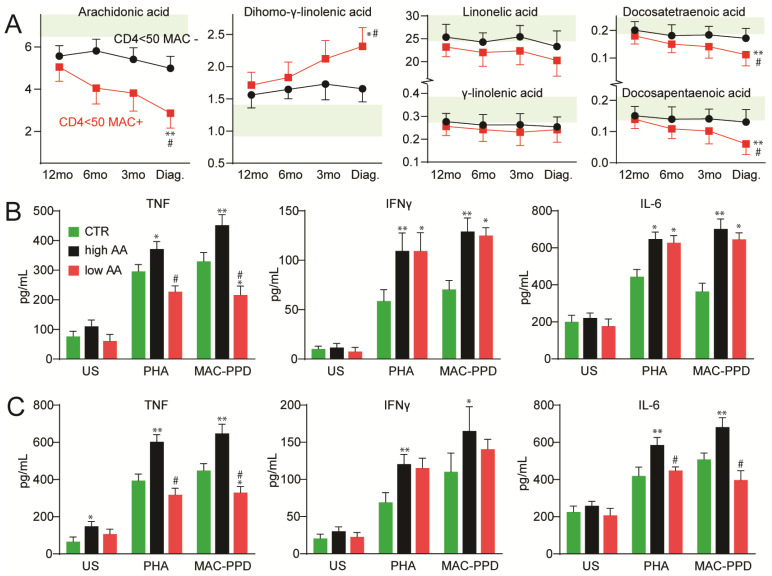
In vivo levels of n-6 PUFAs and ex vivo inflammatory response to MAC infection in HIV-infected patients. Plasma levels of (**A**) n-6 PUFAs arachidonic acid, dihomo-γ-linolenic acid, linoleic acid, γ- linolenic acid, docosatetraenoic acid and docosapentaenoic acid during temporal assessment in HIV-infected patients with CD4+ T cell count <50 cells/µL prior to infection with Mycobacterium avium complex (MAC) (*n* = 31) and at similar times in patients not developing MAC infection (*n* = 28). For A, paired differences within groups were first assessed with the Friedman test for repeated measures followed by the Wilcoxon signed-rank test for evaluating each time point with baseline while between-group differences were compared using the Mann–Whitney U test. * *p* < 0.05, ** *p* < 0.01 vs. baseline, # *p* < 0.05 vs. patients without MAC infection. Green shading indicates the mean ± SD levels in healthy controls. The release of tumor necrosis factor (TNF), interferon (IFN)γ and interleukin (IL)-6 in (**B**) peripheral mononuclear cells and (**C**) bone marrow mononuclear cells from HIV-infected patients with high plasma levels of arachidonic acid (AA) above 7.0 wt% (high AA, PBMC: *n* = 7, BMMC: *n* = 4) and below 5.0 wt% (low AA, PBMC: *n* = 5, BMMC: *n* = 3) and in healthy controls (PBMC: *n* = 7 and BMMC: *n* = 5). The panel shows spontaneous secretion (unstimulated) and secretion after stimulation with phytohaemagglutinin (PHA, dilution 1:200) and purified protein derivate of Mycobacterium avium complex (MAC-PPD, dilution 1:100). Data are given as mean ± SEM. For (**B**,**C**), cytokine levels during stimulation (i.e., PHA or MAC-PPD) were compared to those in unstimulated cells using Wilcoxon signed-rank test while differences between groups (i.e., controls and patients with high or low baseline AA) were compared using Kruskal–Wallis test followed by Mann–Whitney U test to determine the differences between groups. * *p* < 0.05, ** *p* < 0.01 vs. healthy controls. # *p* < 0.05 vs. patients with high levels of AA.

**Table 1 viruses-15-01613-t001:** Overview of Fatty Acids (FA) and cytokines assessed in the study.

	Abbreviation	Lipid Numbers	Method
Saturated FA	SFA		Gas chromatography following chloroform and methanol extraction
Monounsaturated FA	MUFA		
n-3 polyunsaturated FA	n-3 PUFA		
n-6 polyunsaturated FA	n-6PUFA		
dihomo-γ-linolenic acid	DGLA	C20:3n-6	
linoleic acid	LA	C18:2n-6	
γ-linolenic acid	GLA	C18:3n-6	
arachidonic acid	AA	C20:4n-6	
docosatetraenoic acid	DTA	C22:4n-6	
docosapentaenoic acid	DPA	C22:5n-6	
n-9 polyunsaturated FA	n-9 PUFA		
Cytokines			
Tumor necrosis factor	TNF		Enzyme immunoassay
Interferon γ	IFNγ		
Interleukin-6	IL-6		

## Data Availability

Raw data will be made available upon request.

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
