# Peer review of "Fatty Acids Composition and HIV Infection: Altered Levels of n-6 Polyunsaturated Fatty Acids Are Associated with Disease Progression"

_viruses, 2023, doi:10.3390/v15071613_

Round 1

Reviewer 1 Report

Ueland and coll have undertaken a study to demonstrate that  the altered levels of n-6 polyunsaturated fatty acids are correlated to disease progression.  This was demonstrated ex vivo through both cross sectional and longitudinal studies. In addition authors have shown that levels of FAs are predictive in MAC  coinfection in HIV infected patients. The paper is interesting and I would like some clarification from the authors.

Major points

1)      In the longitudinal studies the authors  have shown a correlation between FAs level and progression of the diseases. They found differences  between progressor and non progressor. I would like to know whether the HIV infected individuals were under ART. It seems strange to me that these individuals were just under observation and no therapy was administrated.

2)      Furthermore ART seems not to influence the level of FAs and therefore the viremia is independent on the level of FAs. In other word inhibition of viremia following ART does not affect the plasma level of FAs. If  virus inhibition in individuals under  ART  does not influence the level of FAs, why FAs level increase in HIV infected individuals progressor therefore  with viremia.

3)       How can the authors explain the importance of monitoring FAs in  HIV infected individuals to ameliorate the chronic inflammation and the response to traditional therapy.

Minor points

The authors should revise grammar and English form. Some details such as the raise to power in material and methods must be corrected (not 106 but 10 6)  

Author Response

Reviewer 1

Ueland and coll have undertaken a study to demonstrate that  the altered levels of n-6 polyunsaturated fatty acids are correlated to disease progression.  This was demonstrated ex vivo through both cross sectional and longitudinal studies. In addition authors have shown that levels of FAs are predictive in MAC  coinfection in HIV infected patients. The paper is interesting and I would like some clarification from the authors.

We thank the Reviewer for thorough and thoughtful evaluation of the manuscript as well as for the critical and helpful comments. We respond here in detail to each of the Reviewer’s comments.

Major points

  • In the longitudinal studies, the authors have shown a correlation between FAs level and progression of the diseases. They found differences between progressor and non-progressor. I would like to know whether the HIV infected individuals were under ART. It seems strange to me that these individuals were just under observation and no therapy was administrated.

We apologize for some inconsistency in relation to the different study populations. As stated in the manuscript, samples were collected in the period 1990-2005 and it has now been clearly stated that in the sub-study of progressors and non-progressors, blood samples were collected before ART was available in Norway (June 1996). In addition, we have now included a flow chart fgure to make it easier to understand what patients that were used in which analysis.

  • Furthermore ART seems not to influence the level of FAs and therefore the viremia is independent on the level of FAs. In other word inhibition of viremia following ART does not affect the plasma level of FAs. If virus inhibition in individuals under  ART  does not influence the level of FAs, why FAs level increase in HIV infected individuals progressor therefore  with viremia.

This is a very interesting point and in retrospect, we see that some of the sentences in relation to our interpretation of the relation between viral load and n-6 PUFA was at least somewhat unclear. Thus, in patients with low AA levels and high DGLA levels at baseline, a pattern that was seen in patients with the most severe disease (CDC group C), there was an increase in AA levels and a decrease in DGLA levels during ART. Although not fully normalized, this must be regarded as a potential beneficial effect of ART, and notably, in these patients, but not in the ART group as a whole, we found significant correlations between the change of AA (r=-0.63, p<0.01) and DGLA (r=0.45, p<0.05) and the change in HIV RNA during ART. Thus, at least in a subgroup of patients with the most disturbed AA levels at baseline, there was indication of a possible link between viral load and some n-6 PUFAs (i.e., AA and DGLA). However, ART did not restore AA or DGLA levels to normal levels, even in virally fully suppressed patents, suggesting that other factors than HIV viremia contributes to reduced levels of AA in HIV-infected patients such as persistent low-grade inflammation that may be seen even during successful ART. In the revised manuscript, this part of the text has been partly re-written and this important issue is more thoroughly discussed.

  • How can the authors explain the importance of monitoring FAs in HIV infected individuals to ameliorate the chronic inflammation and the response to traditional therapy

So far measurements of PUFAs are not part of the routine analyses at most hospital so until such analyses are more available on a routine basis, we will not recommend using PUFA measurements in the routine management of people living with HIV. However, supplemental of PUFAs, not only n-3 PUFAs but also n-6 PUFAs, could be considered at least in subgroups such as those with metabolic and cardiovascular co-morbidities. Most importantly, however, we need large randomized controlled trials to evaluate if such supplementation could modulate the low-grade inflammation and immune activation that are seen in people living with HIV even during successful ART. This important issue, that was raised by the Reviewer, is now discussed in the revised manuscript.

Minor points

The authors should revise grammar and English form. Some details such as the raise to power in material and methods must be corrected (not 106 but 10 6)

We have carefully corrected the manuscript to ensure correct grammar and English including correction to 106.   

Reviewer 2 Report

This manuscript examines the role in inflammation and immune regulation based on the hypothesis that a dysregulated FA metabolism may be involved in disease progression in HIV-infected patients. FAs in plasma from HIV infected patients classified as rapid clinical progressors, patients with advanced HIV-related disease acquiring Mycobacterium avium complex (MAC) infection, and also examined if ART influenced the FA pattern in HIV-infected patients.

Introduction:

The introduction provides some detail about the potential role of FAs in inflammatory and metabolic processes and some detail about the FAs chosen for assessment. This section could be focused more on HIV or inflammatory processes but is fine as long as some literature/detail is provided for the rationale of selection for the FAs in this study.

Materials and Methods

Please include a study flow diagram of what patients were used in which analysis to help interpret the results. Which portion of patients was the ART study conducted with? There are 4 separate studies/analysis mentioned here and that is not evident from the hypothesis; please amend this.  Subheadings would help the readability, flow and structure of this manuscript for the 3 separate portions assessed. Line 86-87: Why was such a low volume of age and sex matched controls used? Typically, a higher volume of controls is needed to increase the statistical power. This is an unusual methodology.

It would be helpful to have a table to understand all of the FAs and biomarkers assessed in the different tests as well as a method of assessment.

Line (120) There are 10 healthy controls mentioned here and in the cross sectional analysis 20 healthy controls are mentioned. Are these the same patients?

Statistical Analysis

Please specify which analysis were used for which statistical tests. There were so many different analysis done it’s hard to know whether what was done was appropriate or not without specifying this.

Results

(Line 159) I don’t understand how a stepwise analysis was conducted-none of the designs mentioned used a regression analysis. Please clarify the meaning of the use of the term ‘Stepwise’

Please separate the results of the study into 4 separate sections with subheadings based on the 4 different tests conducted-not the findings of those tests. It’s very confusing to read through the results otherwise and know what was done and the findings.

Conclusion

This section seems well written and appropriate given the findings.

Author Response

Reviewer 2

This manuscript examines the role in inflammation and immune regulation based on the hypothesis that a dysregulated FA metabolism may be involved in disease progression in HIV-infected patients. FAs in plasma from HIV infected patients classified as rapid clinical progressors, patients with advanced HIV-related disease acquiring Mycobacterium avium complex (MAC) infection, and also examined if ART influenced the FA pattern in HIV-infected patients.

We thank the Reviewer for thorough and thoughtful evaluation of the manuscript as well as for the critical and helpful comments. We respond here in detail to each of the Reviewer’s comments

Introduction:

The introduction provides some detail about the potential role of FAs in inflammatory and metabolic processes and some detail about the FAs chosen for assessment. This section could be focused more on HIV or inflammatory processes but is fine as long as some literature/detail is provided for the rationale of selection for the FAs in this study.

Based on the Reviewer´s valuable comment we have expanded the Introduction and included some additional references in relation to the interaction between HIV, inflammation and fatty acids, being the basis for rationale for the present study.  

Materials and Methods

Please include a study flow diagram of what patients were used in which analysis to help interpret the results. Which portion of patients was the ART study conducted with? There are 4 separate studies/analysis mentioned here and that is not evident from the hypothesis; please amend this.  Subheadings would help the readability, flow and structure of this manuscript for the 3 separate portions assessed. Line 86-87: Why was such a low volume of age and sex matched controls used? Typically, a higher volume of controls is needed to increase the statistical power. This is an unusual methodology.

We apologize for not being clearer in the presentation of the different sub-studies and the participation of the different patients in these sub-studies/sub-analyses. Based on the Reviewer´s excellent suggestion, we have now included a flow chart that will help the readability of this part of the manuscript.  We have also changed the sub-headings according to the Reviewer´s suggestion. Finally, although the controls were sex and age-matched in relation to the patients, we agree that the number (n=20) is low and this has been pointed out as a limitation of the study.

It would be helpful to have a table to understand all of the FAs and biomarkers assessed in the different tests as well as a method of assessment.

Again, this is a very good suggestion and we have now included a Table that lists the most important FAs and cytokines in the present study, showing abbreviations, lipid numbers and method of detection.

Line (120) There are 10 healthy controls mentioned here and in the cross sectional analysis 20 healthy controls are mentioned. Are these the same patients?

In the sub-study of PBMC and bone-marrow derived mononuclear cells fewer controls were included (at least partly caused by more invasive procedure), but still the number is low (see above). The sub-study with PBMC and bone-marrow derived mononuclear cells is a particular sub-study separate from the other sub-studies and this has now been clearly stated in the revised manuscript. See also above in relation to inclusion of a flow chart.

Statistical Analysis

Please specify which analysis were used for which statistical tests. There were so many different analysis done it’s hard to know whether what was done was appropriate or not without specifying this.

In the revised manuscript, we have made the “statistical analysis” section clearer, specifying which tests were used for the different study populations. The statistical tests that used for the different analyses have been included in the Figure legends/text where appropriate. 

Results

(Line 159) I don’t understand how a stepwise analysis was conducted-none of the designs mentioned used a regression analysis. Please clarify the meaning of the use of the term ‘Stepwise’

There was no regression analysis. It was meant to indicate an incremental increase with CDC classification but is unnecessary, and this poor phrasing has been removed in the revision.

Please separate the results of the study into 4 separate sections with subheadings based on the 4 different tests conducted-not the findings of those tests. It’s very confusing to read through the results otherwise and know what was done and the findings.

As mention above, we have now separated the Results into separate sections and we have changed the subheading according to the Reviewer´s suggestion.

Conclusion

This section seems well written and appropriate given the findings.

We are glad to know that the Reviewer found this section appropriate

Round 2

Reviewer 2 Report

Thank you very much for your thorough updates. I think the manuscript has been significantly improved for readibility and I am fine with publication. Nice work!